# L-Carnitine: A New Therapeutic Option for the Prevention of Atrial Fibrillation in Non-Cardiac Surgery—A Single-Group Interventional Pilot Study

**DOI:** 10.3390/jcm13206228

**Published:** 2024-10-18

**Authors:** Yasushige Shingu, Isao Yokota, Tatsuya Kato, Yasuhiro Hida, Kichizo Kaga, Jingwen Gao, Satoru Wakasa

**Affiliations:** 1Department of Cardiovascular Surgery, Faculty of Medicine and Graduate School of Medicine, Hokkaido University, Sapporo 060-8638, Japan; 2Department of Biostatistics, Graduate School of Medicine, Hokkaido University, Sapporo 060-8638, Japan; 3Department of Thoracic Surgery, Hokkaido University Hospital, Sapporo 060-8648, Japan; 4Department of Advanced Robotic and Endoscopic Surgery, Fujita Health University, Toyoake 470-1192, Japan; 5Department of Thoracic Surgery, Tonan Hospital, Sapporo 060-0004, Japan

**Keywords:** lung cancer, postoperative atrial fibrillation, L-carnitine, fatty acid-binding protein 4

## Abstract

**Background:** L-carnitine is essential in lipid metabolism and reportedly has preventive effects for arrhythmia. Our objective was to examine the incidence of postoperative atrial fibrillation (POAF) and changes in serum biomarker levels following perioperative L-carnitine administration in patients with lung cancer. **Methods:** Thirteen patients undergoing a lobectomy with preoperative serum brain natriuretic peptide levels >24 pg/mL were perioperatively administered L-carnitine for 5 days (3 g/3×). Accurate 95% confidence intervals (CI) for POAF incidence were calculated. Serum biomarkers for POAF in lung cancer and target proteins for L-carnitine were evaluated by using open-source data from proteomic analysis. **Results:** The incidence of POAF was 38.5% (95% CI 13.9%–68.4%). Fatty acid-binding protein 4 (FABP4) was selected as a candidate biomarker from 1472, 63, and 26 proteins related to lung cancer, L-carnitine, and AF, respectively. A positive correlation was observed between the predicted POAF incidence rate and preoperative FABP4 levels (Pearson’s r = 0.5183). The mean change in serum FABP4 after L-carnitine administration for 5 days was −2.9 ng/mL (95% CI −4.9 to −0.89 ng/mL). **Conclusions:** The incidence of POAF after a lobectomy was 38.5% after the perioperative administration of L-carnitine for patients at a high risk of POAF. The serum FABP4 level demonstrates potential as a candidate biomarker for POAF prediction.

## 1. Introduction

Postoperative atrial fibrillation (POAF) occurs in 13% to 25% of patients undergoing a lung lobectomy [1,2,3], which can be associated with several complications, including hypotension, heart failure, stroke, and systemic embolization. The following risk factors were reported in a large study from the Society of Thoracic Surgeons involving 13,906 patients: age, male sex, pneumonectomy, and clinical stage > II [3]. Preoperative left ventricular diastolic dysfunction and elevated serum brain natriuretic peptide (BNP) are also reported to be risk factors for POAF, independently of age [2,4,5]. POAF prolongs the hospital stay and, in part, raises medical costs due to the increased use of antiarrhythmic and anticoagulation drugs. Although β-blockers and amiodarone are effective in preventing POAF in patients undergoing thoracic surgery [6,7], they are not commonly used due to side effects including bradycardia, hypotension, and pulmonary fibrosis. Therefore, there is an urgent need for effective measures to prevent POAF with fewer side effects.

L-carnitine, which is derived from an amino acid, acts as a transport molecule for the movement of long-chain fatty acids through the inner mitochondrial membrane, and it is essential in lipid metabolism. While L-carnitine has traditionally been used to treat congenital carnitine deficiency, it is also effective in reducing ventricular arrhythmias following myocardial infarction [8] and POAF after coronary artery bypass surgery [9]. The oral administration of L-carnitine is well tolerated, with no serious side effects, even during the perioperative period [10]. To date, however, there have been no reports describing L-carnitine’s administration to prevent POAF after lung cancer surgery. As such, the present single-group interventional pilot study aimed to determine a 95% confidence interval (CI) for the POAF rate with L-carnitine and to offer foundational data for potential future randomized controlled trials. Therefore, we did not assess the effect of L-carnitine administration by comparing it with a control group that did not receive the treatment. Furthermore, we examined the serum levels of a candidate biomarker and assessed the correlation with the predicted POAF rates.

## 2. Materials and Methods

### 2.1. Subjects and Study Design

This study was an open-label, single-arm interventional trial. Patients who had undergone a lobectomy for lung cancer at Hokkaido University Hospital (Sapporo, Hokkaido, Japan) between December 2021 and April 2023 were considered eligible for inclusion. All tumors were staged in accordance with the pathological tumor/node/metastasis classification (8th edition) of the Union for International Cancer Control [11]. Electrocardiography and echocardiography were performed within 3 months prior to the surgery. Electrocardiography was used to evaluate the heart rhythm, PQ interval, and QRS duration. Transthoracic echocardiography was employed to measure the left ventricular end-diastolic dimension, ejection fraction, and left atrial size. Only patients with preoperative serum BNP levels > 24 pg/mL were included in the study because they were high-risk patients for POAF, based on the authors’ previous study (POAF rate, 31%) [12]. Patients who had a history of atrial fibrillation (AF), life-threatening arrhythmias, unstable angina, acute myocardial infarction, or renal failure requiring hemodialysis or were using digitalis or class I and III antiarrhythmic medications were not included in the study. This study was carried out in accordance with the principles embodied in the Declaration of Helsinki.

### 2.2. Selection of Serum Biomarker

Serum biomarkers for POAF in lung cancer and targets for L-carnitine are unknown. Figure 1 shows the Venn diagram for biomarker selection. We searched the “Cancer Serum Atlas” (www.cancerserumatlas.com; accessed on 19 December 2023) and acquired 1472 protein IDs as the serum biomarkers in lung cancer [13]. The names of the proteins were obtained by using the UniProt database (https://www.uniprot.org/; accessed on 19 December 2023). L-carnitine reportedly activates peroxisome proliferator-activated receptors (PPARs), which are nuclear receptors that regulate lipid and glucose metabolism [14]. For the targets of L-carnitine, 63 PPAR-pathway related proteins were listed in the Kyoto Encyclopedia of Genes and Genomes (KEGG) (https://www.genome.jp/kegg/; accessed on 19 December 2023). In a proteomics study, 26 proteins were identified as risk factors for the new onset of AF [15]. Finally, fatty acid-binding protein 4 (FABP4) was selected as a candidate biomarker related to lung cancer, L-carnitine, and the new onset of AF.

### 2.3. L-Carnitine Administration and Estimation of Serum Biomarker Levels

L-carnitine (L-Cartin FF oral solution 10%; Otsuka Pharmaceutical Co., Ltd., Tokyo, Japan) was administered orally to patients for 2 days before surgery and for 3 days postoperatively. Three oral doses of 1 g each were administered (1833 JPY [12.4 USD]/day). This dose was based on a meta-analysis [8] that indicated that 2 g/day was the minimally effective dose to prevent arrhythmia. Furthermore, the maximum dose was 3 g/day according to the company’s product labels.

Blood samples were collected at 10:00 and 06:00 on the first and last days of L-carnitine administration, respectively. After centrifugation (4 °C) at 3500 rpm for 10 min, the serum was preserved at −80 °C until further analysis. The FABP4 level was estimated using a commercially available kit (Human FABP4/A-FABP ELISA; ThermoFisher Scientific Inc., Waltham, MA, USA), in accordance with the manufacturer’s instructions.

### 2.4. Definition of POAF and Monitoring

The occurrence of POAF was tracked through a 24 h electrocardiogram over the first three days post-surgery. POAF was characterized by AF episodes lasting more than 30 s [16]. The diagnosis of POAF was determined using the following criteria observed in leads II, III, or aVF: the absence of P-waves, the presence of irregular and fine baseline tremors, and irregular RR intervals. The diagnosis of POAF was validated by an investigator (Y.S.) and a cardiac surgeon who were not involved in this study.

### 2.5. Predicted POAF Rate

The predicted POAF rate was determined using an application developed by Amar et al. [5], using the age, body mass index, history of AF, and preoperative BNP level. The developer provided the application. Plasma BNP levels were estimated within 1 month of surgery.

### 2.6. Statistical Analysis

The confidence level was set at 95%. Data for continuous variables were expressed as the median (interquartile range (IQR)). A sample size of 13 was necessary to ensure that the width of the 95% confidence interval was less than 40%. The incidence of POAF within 3 days of surgery was calculated using an exact Clopper–Pearson 95% CI using EZR (Jichi Medical University Saitama Medical Center, Saitama, Japan) for R. The mean change in the serum level of FABP4 was estimated, and these 95% CIs were constructed by t-distribution. The relationship between the preoperative FABP4 levels and anticipated rates of postoperative atrial fibrillation (POAF) was assessed using Pearson’s correlation coefficient (r). The analysis was conducted using GraphPad Prism version 9.4.1 (GraphPad Inc., San Diego, CA, USA).

## 3. Results

### 3.1. Preoperative Data

Although 14 patients were enrolled, ECG monitoring was accidentally discontinued in one; as such, 13 subjects who completed the protocol were included in this study. The patients’ preoperative characteristics are summarized in Table 1. The patients’ age was 75 years (IQR 68–79 years). Most patients had dyslipidemia and were in the early clinical stages of lung cancer. Only one patient received β-blockers preoperatively. The median BNP value was 36.5 pg/mL (IQR 26.6–62.7 pg/mL).

The preoperative cardiac function parameters are summarized in Table 2. All patients exhibited normal sinus rhythms, and their median heart rate was 73 beats/min (IQR 61–83 beats/min) before surgery. All patients exhibited a normal left ventricular size and ejection fraction. The mean left atrial diameter was 36 mm (IQR 29–39 mm). Moderate aortic valve and tricuspid valve regurgitation were observed in one patient each.

### 3.2. Operative and Postoperative Data

The surgical and postoperative data are presented in Table 3. Robotic surgery was carried out on three patients, while the remaining underwent video-assisted thoracoscopic surgery. The cancer pathologies encountered in the study included squamous cell carcinoma (*n* = 5 [38%]) and adenocarcinoma (*n* = 8 [62%]). There were no adverse effects associated with the administration of L-carnitine, such as loose stools, diarrhea, flushing, or low blood sugar levels, and the study protocol was completed by all 13 patients. No postoperative morbidity or mortality was observed.

### 3.3. Incidence of POAF

POAF (>30 s) was observed in five patients, corresponding to an incidence of 38.5% (95% CI 13.9%–68.4%). The durations of POAF were 14 h, 5.5 h, 10 min (twice), 1 min, and 30 s (multiple times). All episodes of POAF occurred on postoperative days 2 or 3. Anticoagulation therapy and β-blocker administration were required due to the recurrence of AF during hospitalization in one patient. None of the patients experienced atrial flutter. The median duration of the hospital stay after surgery was greater for patients who had POAF compared to those who did not experience it: 8.0 days (IQR 6.5–9.0 days) versus 6.0 days (IQR 5.0–6.5 days), respectively.

### 3.4. Serum FABP4 Levels

The predicted rate of POAF based on the preoperative FABP4 levels is shown in Figure 2a. A significant association was found between the preoperative FABP4 level and the estimated risk of developing POAF (r = 0.5183). The changes in the serum FABP4 levels after L-carnitine administration are shown in Figure 2b. The mean serum FABP4 level decreased from 12.4 to 9.5 ng/mL (mean change, −2.9 [95% CI −4.9 to −0.89 ng/mL]). The median preoperative FABP4 levels in POAF and non-POAF patients were 13.8 (IQR 11.0–15.7) vs. 11.4 (IQR 9.1–14.2) ng/mL, respectively.

## 4. Discussion

The perioperative oral intake of L-carnitine in patients who underwent a lobectomy was safe and well tolerated. The incidence of POAF (>30 s) after the lobectomy was 38.5% after the perioperative administration of L-carnitine in high-risk patients for POAF. Furthermore, a correlation was suggested between the preoperative level of FABP4 and the POAF risk score.

### 4.1. L-Carnitine for Prevention of POAF

A randomized study involving 134 patients who underwent coronary artery bypass grafting indicated that the perioperative administration of L-carnitine (3 g/day) decreased the incidence of POAF from 19.4% to 7.5% [9]. However, no studies have used L-carnitine in lung surgery. In the current study using L-carnitine, the incidence of POAF (>30 s) was 38.5% in high-risk patients with elevated BNP levels. When we reanalyzed the incidence of POAF lasting >5 min, as in previous investigations [1,5,12], the POAF rate was 23.1% (3/13 [95% CI 5.0–53.8%]). The POAF rate has been reported to be >25% in elderly patients undergoing a lobectomy [1]. The rate was reportedly 36.8% in patients with a low left ventricular ejection fraction [17]. Furthermore, the incidence of POAF was 31% in our previous retrospective study, which included a similar population that was not administered L-carnitine [12]. Future randomized controlled studies with control groups are necessary to examine the clinical benefits of L-carnitine in preventing POAF.

### 4.2. FABP4 as Biomarker for POAF in Lung Cancer

The results of this study indicated a positive correlation between the preoperative FABP4 levels and the predicted rate of POAF. A calculator for the POAF risk has been developed in the United States [5], although it has not yet been validated in a cohort of Japanese patients; however, the correlation is not a mere coincidence.

FABP4 was selected in the workflow using an open-source database related to lung cancer, L-carnitine (PPAR pathway), and AF (Figure 1). Tissue FABP4 is a factor for poor prognosis in lung cancer; the high gene expression of FABP4 in tissue predicts survival in patients with squamous cell carcinoma [18]. Furthermore, FABP4 is released from cancer-associated adipocytes after lipolysis induction and has an important role in tumor initiation and progression [19]. In two independent community-based cohorts (total 3338 cases) in Sweden, Lind L. and colleagues identified novel markers (FABP4, IL-6, and TIM-1) for the new onset of AF via a proteomics analysis of 85 proteins [15]. In another previous study that analyzed the plasma levels of several adipokines in patients undergoing AF ablation, the FABP4 level was the best predictor of AF recurrence after catheter ablation [20]. Further studies with larger sample sizes are necessary to confirm the utility of FABP4 as a predictor of POAF in lung cancer surgery.

### 4.3. Decrease in Level of Serum FABP4 after L-Carnitine

The significant reduction in serum FABP4 levels after L-carnitine administration (mean change, −2.9 ng/mL [95% CI −4.9 to −0.89 ng/mL]) merits some discussion. Although no studies have examined the alterations in serum FABP4 levels following L-carnitine administration, there was evidence that L-carnitine inhibited FABP4 gene expression in a model of burn-induced cellular damage in hepatocytes [21]. While the exact mechanism is still unknown, L-carnitine may reduce the levels of FABP4.

It is possible that tumor resection―but not L-carnitine administration―reduces the serum FABP4 levels. The FABP4 expression in the adjacent normal tissue is reported to be high in various malignant tumors, including lung cancer, compared with that in tumor tissue [22]. If the cancer-associated adipocytes that release FABP4 are included in the adjacent normal tissue, tumor resection might reduce the serum FABP4 levels. To address this issue, we need to examine the changes in the FABP4 levels after lung surgery without L-carnitine in future studies. Finally, while the FABP4 levels decreased in all patients who experienced POAF, the levels increased in the two cases without POAF (Figure 2b). Further studies are needed to determine the importance of the postoperative FABP4 levels.

### 4.4. Limitations

This study had several limitations, the most notable being that the observation period for POAF lasted only three days following surgery. While POAF typically occurs within this timeframe, it is possible that the incidence of POAF during hospitalization was underestimated. Second, we did not assess the blood levels of free carnitine. In our previous study, we established that the serum free carnitine levels increased twofold following the 9-day oral administration of L-carnitine in patients undergoing cardiac surgery [10]. Although we speculated that the carnitine concentrations were significantly elevated in the current study, the serum levels may have varied between the patients. Third, we did not include a control group without L-carnitine for comparison. A future randomized study is necessary, using the findings from the current study.

## 5. Conclusions

The incidence of POAF (>30 s) was 38.5% after the perioperative oral administration of L-carnitine during a lobectomy for lung cancer in high-risk patients for POAF. The preoperative serum FABP4 level could be a candidate marker for POAF prediction. The findings of this study could help to guide the determination of sample sizes for future randomized trials.

## Figures and Tables

**Figure 1 jcm-13-06228-f001:**
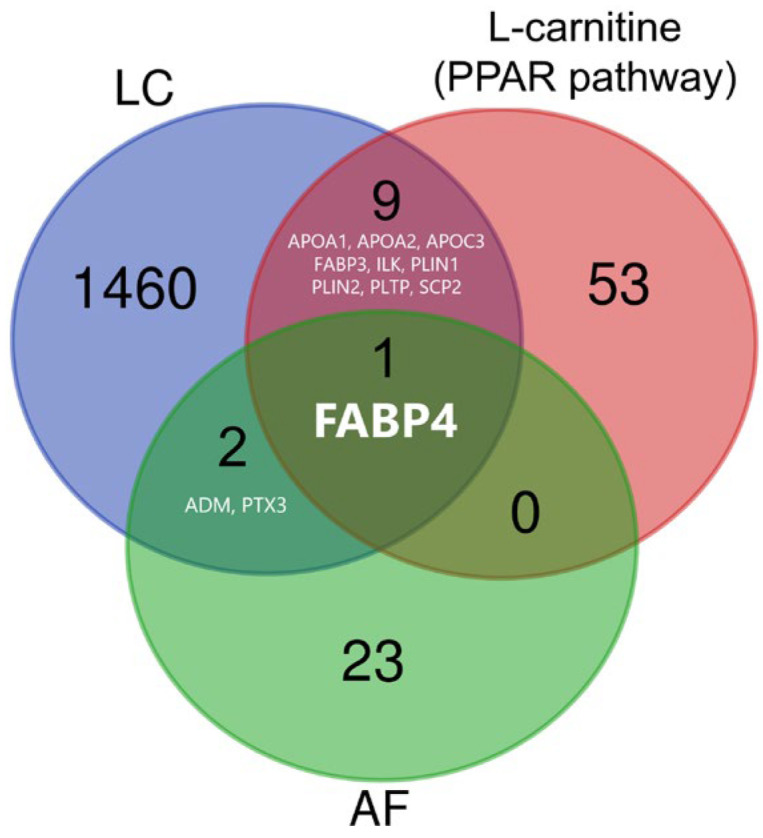
Venn diagram for the selection of serum biomarkers. ADM, adrenomedullin; AF, atrial fibrillation; APO, apolipoprotein; FABP, fatty acid-binding protein; ILK, integrin-linked kinase; LC, lung cancer; PLIN, perilipin; PLTP, phospholipid transfer protein; PPAR, peroxisome proliferator-activated receptor; PTX3, pentraxin 3; SCP, sterol carrier protein.

**Figure 2 jcm-13-06228-f002:**
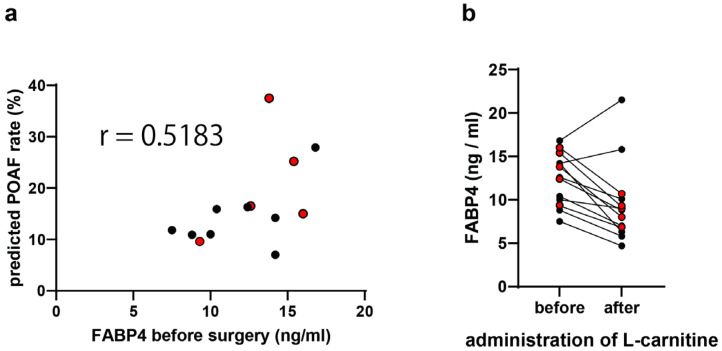
Serum FABP4 levels. (**a**) Correlation between FABP4 and the predicted POAF rate. (**b**) Changes in FABP4 after L-carnitine administration. The red markers indicate the patients with POAF (>30 s). FABP4, fatty acid-binding protein 4; POAF, postoperative atrial fibrillation.

**Table 1 jcm-13-06228-t001:** Preoperative characteristics.

Variable	*n* = 13
Male, *n* (%)	7 (54)
Age, yrs, median (IQR)	75 (68, 79)
BMI, kg/m^2^, median (IQR)	22.9 (18.6, 24.5)
BNP, pg/mL, median (IQR)	36.5 (26.6, 62.7)
Respiratory function	
VC, L	3.3 (2.8, 4.2)
FVC, L	3.3 (2.7, 4.0)
FEV1.0, L	2.4 (2.1, 2.9)
FEV1.0/VC, %	75 (71, 84)
cStage, *n* (%)	
IA/IB	9 (69)/1 (8)
IIA/IIB	1 (8)/1 (8)
IIIA	1 (8)
Diabetes mellitus, *n* (%)	4 (31)
Hypertension, *n* (%)	6 (46)
Dyslipidemia, *n* (%)	9 (69)
Medications	
β-Blockers, *n* (%)	1 (8)
Statins, *n* (%)	5 (38)

BMI, body mass index; BNP, brain natriuretic peptide; FEV, forced expiratory volume; FVC, forced vital capacity; IQR, interquartile range; VC, vital capacity.

**Table 2 jcm-13-06228-t002:** Preoperative cardiac functional parameters.

Variable	*n* = 13
Systolic BP, mm Hg, median (IQR)	127 (109, 146)
Diastolic BP, mm Hg, median (IQR)	69 (68, 80)
Electrocardiogram	
Heart rate, bpm, median (IQR)	73 (61, 83)
PQ, ms, median (IQR)	155 (142, 181)
QRS, ms, median (IQR)	102 (93, 109)
Transthoracic echocardiography	
LVDd,	42 (40, 46)
LVEF, %, median (IQR)	68 (63, 73)
Left atrial diameter, mm, median (IQR)	36 (29, 39)

BP, blood pressure; IQR, interquartile range; LVDd, left ventricular end-diastolic dimension; LVEF, left ventricular ejection fraction.

**Table 3 jcm-13-06228-t003:** Operative and postoperative data.

Variable	*n* = 13
Robotic surgery, *n* (%)	3 (23)
Side of lobectomy, *n* (%)	
Left/right	3 (23)/10 (77)
Resected lobe, *n* (%)	
Upper/middle/lower	10 (77)/1 (8)/2 (15)
Lymph node dissection, *n* (%)	
ND1a/ND1b	1 (8)/1 (8)
ND2a-1/ND2a-2	8 (62)/3 (23)
Operative time, min, median (IQR)	195 (162, 238)
Bleeding, mL, median (IQR)	0 (0, 95)
Postoperative hospital stay, days, median (IQR)	6 (5, 8)
Pathology of the tumor, *n* (%)	
Squamous cell carcinoma	5 (38)
Adenocarcinoma	8 (62)
Postoperative complications, *n* (%)	0 (0)
Hospital death, *n* (%)	0 (0)

IQR, interquartile range.

## Data Availability

The data that support the findings will be made available on request by the corresponding author. Restrictions apply to the availability of these data, which were used under license for the current study.

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
