# Peer review of "L-Carnitine: A New Therapeutic Option for the Prevention of Atrial Fibrillation in Non-Cardiac Surgery—A Single-Group Interventional Pilot Study"

_jcm, 2024, doi:10.3390/jcm13206228_

Round 1
Reviewer 1 Report
Comments and Suggestions for Authors
The authors of the current study present results from a small-sample, pilot interventional study investigating the effect of L-carnitine, administered for 5 days pre-operatively to patients who had been planned for elective lobectomy for a lung cancer. Auhtors' hypothesis was that this treatment would reduce the incidence of peri-/postoperative atrial fibrillation and the level of some serum biomarkers like BNP. The manuscript is topical and important from clinical point of view. Atrial fibrillation is the most common sustained arrhythmia in humans, affecting 1 of 3-4 aged >55 years (lifelong risk for at least 1 episode of AF) with even higher incidence among patients with certain comorbidities and surgical procedures. On the other hand, AF is associated with high risk of strokes/arterial thromboembolic events, so any treatment that could prevent AF occurrence will be appreciated by the clinicians and their patients.
I have some comments and recommendations to the authors of this manuscript:
1. The title has to be modified/improved to become more attractive to the readers, for example "L-carnitine: a new therapetic option for AF prevention in non-cardiac surgery" or something like that.
2. A control group is missing and it is very improtant for such an interventional study. How would we know that patients with the same clinical profile who undergo lobectomy for a lung cancer and who are not treated with L-carnitine would have higher incidence of AF and higher levels of the invetigated laboratory parameters?
3. Study limitations should be added at the end of the "Discussion"
Comments on the Quality of English LanguageMinor editing of English language is required.
Author Response
We should like to thank to the reviewer for the constructive criticisms. We have addressed them in the manuscript as well as point by point below.
- The title has to be modified/improved to become more attractive to the readers, for example "L-carnitine: a new therapeutic option for AF prevention in non-cardiac surgery" or something like that.
Authors response:
As the reviewer suggested, we have changed the title to “L-carnitine: a new therapeutic option for the prevention of atrial fibrillation in non-cardiac surgery - A single-group interventional plot study” (Line 2).
- A control group is missing and it is very important for such an interventional study. How would we know that patients with the same clinical profile who undergo lobectomy for a lung cancer and who are not treated with L-carnitine would have higher incidence of AF and higher levels of the investigated laboratory parameters?
Authors response:
We completely agree with the reviewer’s comments. This was a pilot study conducted prior to a future randomized trial. We have revised the title and included this in the new 'Limitations' section as follows: “Third, we did not include a control group without L-carnitine for comparison. A future randomized study is necessary, using the findings from the current study” (Line 269).
- Study limitations should be added at the end of the "Discussion"
Authors response:
As the reviewer suggested, we added the “Limitation” section (Line 261): “The present study had several limitations, the first of which was that the observation period for POAF was only 3 days after surgery. Although POAF usually occurs within this period, the rate of POAF during hospitalization may have been underestimated. Second, we did not measure blood concentrations of free carnitine. We previously confirmed that the level of serum free carnitine doubled after a 9-day oral intake of L-Cartin in patients undergoing cardiac surgery [23]. Although we speculated that carnitine concentrations were significantly elevated in the current study, the serum levels may have varied between patients. Third, we did not include a control group without L-carnitine for comparison. A future randomized study is necessary, using the findings from the current study”.
Reviewer 2 Report
Comments and Suggestions for Authors
The subject is a very interesting one, with some clinical and practical implications
The design of the study is well done
but, some phrases needs to be reformulated, also the title.
the number of the patients enrolled is to small in order to allow pertinent conclusions.
It may be considered a pilot study.
I have inserted few comments directly into the text.
It should be taken into account that the mechanisms by which paroxysmal atrial fibrillation occurs perioperatively in thoracic surgery are much more complex and perhaps should be mentioned in the article

moderate english language revisions
Author Response
We should like to thank to the reviewer for the constructive criticisms. We have addressed them in the manuscript as well as point by point below.
- Some phrases need to be reformulated, also the title.
Authors response:
We appreciate the valuable comments. We have revised the title to: “L-carnitine: A New Therapeutic Option for the Prevention of Atrial Fibrillation in Non-Cardiac Surgery—A Single-Group Interventional Pilot Study”. Additionally, other sections have been corrected based on the reviewers' suggestions. The English has been thoroughly checked and rephrased with the assistance of ChatGPT."
Round 2
Reviewer 1 Report
Comments and Suggestions for Authors
The authors have observed most of the reviewer's recommendations and have made the proposed corrections/additions to their manuscript, which has been significantly improved. It can now be published in its current version.
Reviewer 2 Report
Comments and Suggestions for Authors
the article was improved
please correct in the title - pilot not plot
Comments on the Quality of English Languageminor english language revisions